# OpenReview forum: "MBBQ: A Dataset for Cross-Lingual Comparison of Stereotypes in Generative LLMs"
_colmweb.org/COLM/2024/Conference — COLM_

### Official Review · Reviewer_roCz · 2024-04-15

**Rating:** 8
**Confidence:** 4
**Ethics Flag:** 1

**Summary:**

The paper is about a project that extends the BBQ bias benchmark corpus to "MBBQ", which covers several natural languages. Several open-source or proprietary LLMs are then evaluated multilingually. Results show performance differences between languages, and incidentally, that biases of a LLM differ across bias categories.

**Questions To Authors:**

Some text in the figures and tables is small to the extent of being difficult to read. Make it much larger.

**Reasons To Accept:**

Multilingual bias evaluation of LLMs is appropriate and timely, as much bias work has focused on English or, even if not English, just one language. The multilingual expansion was done with attention to cultural norms, excluding stereotypes that don't travel well. MBBQ will be a useful language resource for bias testing. The results illustrate differences in performance, both across languages and across bias categories.

**Reasons To Reject:**

The paper lacks a discussion section. It's appropriate to devote some space to helping the reader digest results (separately from the Results section) and contextualize them. The findings list in the Conclusion is a step toward that, but it should be expanded upon.

---

> ### Author Rebuttal · Authors · 2024-05-28
>
> Many thanks for dedicating your time and providing feedback! We will use the extra page allowed in the camera ready version to make the figures and tables more readable, and to extend the discussion of our results.In particular, we plan to elaborate on the difference between the results for the two context types, and the relation between accuracy and bias in each context type.

---

### Official Review · Reviewer_eD9a · 2024-05-09

**Rating:** 7
**Confidence:** 3
**Ethics Flag:** 1

**Summary:**

This paper introduces a Multi-lingual Bias Benchmark for Question Answering (MBBQ), an extension of English BBQ dataset to Dutch, Spanish, and Turkish. MBBQ only considers the stereotypes that are commonly held across all these languages to control for the cultural differences. The paper also introduces a control dataset for measuring task performance of the models on question answering in these languages independently of the bias. The authors then carry out evaluation and detailed analysis of 7 chat-optimized LLMs (including Aya, ChatGPT, and LLAMA-2 Chat 7B, etc) on the proposed dataset.

**Reasons To Accept:**

This is a useful benchmark for studying LLM’s biases in non-English langauges.
The writing is generally clear and experiments are well carried out.

**Reasons To Reject:**

Some parts of the paper are unclear and need to be revised:
- The authors mentioned the use of 5 different prompts for eliciting MCQ answers from LLMs but it’s unclear how they were used. Are the results using a single best template after prompt tuning, or average across 5 prompts? It’d be good to clarify this.
- "If no answer can be detected in the model’s response we consider this as neither a biased nor a counter-biased answer." Does this mean these samples are discarded? How would this be used in Equation 1 and 2 (e.g. Correct answer can be detected in biased context but no answer detected in counter-biased context)?

---

> ### Author Rebuttal · Authors · 2024-05-28
>
> Thank you very much for your time and helpful feedback!
>
> Regarding your first question, the results are an average across 5 prompts. We will make sure to clearly state this in section ‘4.1 Models and prompts’ of the paper.
>
> Regarding your second question, instances for which no answer can be detected in the model’s response are indeed discarded when it comes to calculating the bias scores, which means that they are excluded from any of the counts that are part of Equation 1 and 2. As mentioned at the beginning of section ‘5.1 Ability to answer multiple choice questions’, we detect an answer for at least 99% of samples across all prompts and languages for the models for which we compute bias scores. The two models for which we are not able to detect as many answers, Falcon and Wizard, are excluded from the analysis on model biases because of their low accuracy on control-MBBQ. Given the small percentage of non-detected answers (less than 1%), those answers do not have a substantial impact on the reported bias scores calculated according to Equation 1 and 2.

---

### Official Review · Reviewer_B57e · 2024-05-11

**Rating:** 7
**Confidence:** 3
**Ethics Flag:** 1

**Summary:**

This paper introduces a multi-lingual bias evaluation dataset by translating a subset of BBQ examples into Dutch, Spanish and Turkish. They apply machine-translation to the BBQ templates, with some manual human checking along the way (of the templates, of the names going into the placeholders). They report variance in the bias and capability of different language models across different languages.

**Questions To Authors:**

State the date or fully qualified model name for GPT-3.5 Turbo

**Reasons To Accept:**

- The construction of the dataset is reasonable and follows the template established by prior work.
- Evaluations are performend on a sufficiently representative set of modern models.

**Reasons To Reject:**

- The dataset only spans a small set of languages (Dutch, Spanish, Turkish and English), which would be insufficient to measure bias in a broader multi-lingual setting.
- Templates are machine-translated - while they are manually checked, it is not clear if they would be as effective/informative if they were all natively written

---

> ### Author Rebuttal · Authors · 2024-05-28
>
> We appreciate your time and valuable input!
>
> First, we agree that more languages are required to make a full multilingual comparison, and hope that our work describing the process of obtaining translations in the languages we cover enables others to repeat the process in other languages.
>
> Second, we agree that manual translations are generally preferable, but this was not feasible for the scale of our dataset. Therefore, we chose to systematically have the automatic translations checked by native speakers. In particular, we asked them to check all prompts and answer detection phrases, and those translations for which the Google Translate API and the NLLB model differed. This resulted in 90-99% of the context templates, 50-92% of the questions, and 35-89% of (often single word) target groups and terms used for template diversity being manually checked, depending on the language.
>
> Third, we will make sure to include the date for GPT3.5-Turbo.

---

### Decision · Program_Chairs · 2024-07-10

**Decision:**

Accept

**Comment:**

**Paper Summary:**

This paper presents MBBQ (Multi-lingual Bias Benchmark for Question-Answering), which is curated and translated from the English BBQ dataset to Dutch, Spanish, and Turkish. This dataset is curated to contain only common stereotypes across these languages (and cultures). With the dataset, the experiments show that the current 7 LLMs have a severe bias in some non-English than one in English and cross-lingual differences in bias.

**Summary of Strengths:**

- MBBQ is a valuable and timely dataset to investigate social bias in LLMs in non-English languages.
- Reasonable and soundly construction of the dataset and evaluation processes.
- Evaluation results across several LLMs underscore their performance gaps in non-English languages.

**Summary of Weakness:**

- The dataset only spans a small set of languages (Dutch, Spanish, Turkish and English), which would be insufficient to measure bias in a broader multi-lingual setting.
- The current manuscript needs to be improved for the inclusion of some details, including the discussion section and experimental details, which should be modified in the camera-ready version.

Overall, this paper is clearly and soundly written and would be expected to contribute to the community.